# CALC-X and CALCFORMERS: Empowering Arithmetical Chain-of-Thought through Interaction with Symbolic Systems

**Marek Kadlčík**[◇][*] and **Michal Štefánik**[◇][*] and **Ondřej Sotolář**[◇] and **Vlastimil Martinek**[♣]

[◇]Faculty of Informatics, Masaryk University, Czech Republic

[♣]The Central European Institute of Technology, Masaryk University, Czech Republic

{kadlcik,stefanik.m,xsotolar,445261}@mail.muni.cz

## Abstract

Despite outstanding performance in many tasks, language models are notoriously inclined to make factual errors in tasks requiring arithmetic computation. We address this deficiency by creating CALC-X, a collection of datasets that demonstrates the appropriate use of a calculator in reasoning chains. CALC-X is suitable for teaching language models to offload computations to a symbolic system. We survey and unify several existing chain-of-thought datasets into a proposed format, resulting in a standard collection of over 300,000 samples requiring arithmetic reasoning. Finally, we use the new CALC-X collection to train open-source calculator-using models we call CALCFORMERS and show that these models approximately double the accuracy of generating correct results compared to vanilla language model baselines. We make all CALC-X datasets, source code and CALCFORMERS models publicly available.[1]

## 1 Introduction

While the language models (LMs) demonstrate outstanding efficiency in working with unstructured language data, they struggle with problems that require exact computations (Patel et al., 2021b). On the other hand, symbolic systems, such as a calculator, can perform arithmetics without errors. Thus, combining the strengths of both neural and symbolic systems can yield significant benefits in tackling tasks that require arithmetics (Schick et al., 2023; Gao et al., 2023).

Given a sufficient amount of supervised data, the interaction with symbolic systems can be learned. However, obtaining texts demonstrating the interaction in relevant situations and in a consistent structure is non-trivial. Consequently, a substantial effort of most related work addresses the data scarcity problem through semi-supervised learning, heuristics, prompting or few-shot, and reinforcement-

---

*Equal contribution

[1]https://github.com/prompteus/calc-x

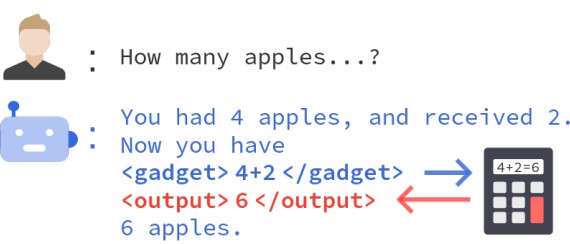

Figure 1: **Generation process of CALCFORMER models**: By generating the closing *</gadget>* tag, model calls an external tool. The following tokens are inserted into the model's context by the tool. Finally, the model continues generation using all of the previous tokens.

based approaches (Section 2) with compromises in the quality and reproducibility.

To support future research in developing open-source tool-assisted language models, we curate a CALC-X collection of over 300,000 samples for mathematical reasoning. CALC-X transforms several existing datasets into a unified format that can be used to train and evaluate LMs for the correct use of a calculator. We survey existing chain-of-thought (CoT) datasets for arithmetical reasoning (Section 2) and pick a subset suitable for integration into a consistent collection. To enable efficient integration of LMs with independent tools, we propose a unified format of a fully parseable HTML-like markup language (Section 3). For each dataset, we describe the curation process of its calculator-augmented (CALC-X) version (Section 4). Finally, we show that training on a full mixture of CALC-X datasets enables LMs to use a symbolic system during inference and largely improves the accuracy on held-out math problems (Section 5). We make all our building tools, datasets and models publicly available (Appendix A.4).

## 2 Related Work

**Math Datasets** *GSM8K* (Cobbe et al., 2021) contains grade-school math problems with human-written CoT explanations and explicit annotation of formulas. *ASDiv* (Miao et al., 2020), SVAMP (Pa-

tel et al., 2021a), MAWPS (Koncel-Kedziorski et al., 2016) and larger Chinese *Ape210k* (Zhao et al., 2020) contain problems of similar complexity, but the solutions are written as nested expressions. *AQuA-RAT* (Ling et al., 2017) contains multiple-choice problems with selected answers and free-text rationales. *MathQA* (Amini et al., 2019) is a subset of *AQuA-RAT* with additional annotation: nested expressions that lead to an answer, similar to those in *Ape210k* or *ASDiv*. *MATH* and *AMPS* (Hendrycks et al., 2021) consist of more challenging problems and contain CoT solutions formatted in LaTeX. Specifically, *MATH* is a set of high-school math competition problems, and *AMPS* is a large-scale pre-training dataset, partly scraped and partly synthetically generated. *Mathematics Dataset* (Saxton et al., 2019) is another generated dataset but containing only final answers.

**Tool-using LMs** A main contribution of much of the previous work in building tool-using models addresses the problem of data scarcity. Komeili et al. (2022), *WebGPT* (Nakano et al., 2021), and *LaMDA* (Thoppilan et al., 2022) let crowd workers annotate tool calls to train models to use a web search, a calculator, and a machine translation system. *PAL* (Gao et al., 2023) applies prompt engineering to make an LM use a Python interpreter without training. *Toolformer*'s approach (Schick et al., 2023) is to prompt an LM to insert gadget tags ("API calls") into CoT datasets and filter out irrelevant ones using the trained model's perplexity. In evaluation, Toolformer simplifies the problem to only one tool call per example and supports generation with several heuristic rules.

*TaLM* (Parisi et al., 2022) extends a training dataset by *self-training*: They start with a small set of fully annotated data, including the annotations of tool calls, and then iteratively generate CoTs with tool calls for a larger dataset with incomplete annotation. The examples added to the training set are chosen heuristically without guarantee that the entire reasoning chain is correct.

We note that *none* of the referenced work publicly releases the resulting models. In combination with the many training and inference heuristics, it is largely difficult to reproduce and build upon the proposed methods. However, the availability of an extensive, standardized collection of tool-assisted datasets like the one presented by CALC-X will allow future work to substantially simplify the methods needed for creating tool-assisted models.

# 3 CALC-X Interaction Format

We propose a semi-structured format for CoT data to provide both the flexibility of unstructured text and the precision of structured formats. The HTML-based structure of interactions is compatible with existing parsers, such as Beautiful-Soup (Richardson, 2007). This allows fast execution of parsing in the interaction with tools within the generation but also allows future work to easily transfer CALC-X collection into other desired interaction formats.

Our format, displayed in Figure 2, uses three tags: *gadget*, *output*, and *result*. Tag *gadget* is intended for inputs or "queries" to an external system. Tag *output* wraps the response of the external system to the query. The tag *result* wraps the final result of the thought chain.

```
After buying the bread and candy bar,
you have 32-3-2=
<gadget id="calculator">32-3-2</gadget>
<output>27</output>
$27. You spend 27/3=
<gadget id="calculator">27/3</gadget>
<output>9</output>
9 dollars on the turkey. You have 27-9=
<gadget id="calculator">27-9</gadget>
<output>18</output>
$18 left. The final result is 18.
<result>18</result>
```

Figure 2: An example of target text from a chain-of-thought dataset encoded in our proposed format. Our format is designed to allow the interaction of LMs with multiple external systems, such as a calculator.

# 4 Creation of CALC-X Collection

Out of the datasets reviewed in Section 2, we create the first version of CALC-X collection from these datasets: GSM8K, AQuA-RAT, MathQA, Ape210k, MAWPS, SVAMP and ASDiv. Our selection considers the datasets' size, primary focus on arithmetics, and parseability of the tool calls.

The resulting CALC-X collection is designed to simplify the *correct* usability of the whole collection in *both* training and evaluation while persisting the maximum of datasets' original information. Most importantly, the process includes (1) the unification of key attributes (i.e. *inputs*, *labels* and *correct results*) over all datasets, and (2) the elimination of data leakages between different (train-/val/test) data splits throughout the *whole* collection.

We perform the second step based on a lexical overlap between pairs of samples' input texts from different splits across *all* datasets. We consider a

pair of train and test examples a leak if the Jaccard similarity of their 1-gram and 2-gram representations is over 0.5. This results in data splits composed of *subsets* of the original datasets, but thanks to this step, the whole CALC-X collection can be used to perform both validation and tests over *all* datasets when *all* datasets are also used in training.

The remainder of this section describes the conversion process of each dataset currently included in the CALC-X collection.

## 4.1 GSM8K

GSM8K (Cobbe et al., 2021) is a CoT dataset with over 8,000 examples containing arithmetical expressions that can be evaluated using a calculator. The syntax is not standard but can be easily parsed.

"Natalia sold 48/2 = $\langle\langle 48/2 = 24 \rangle\rangle$ 24 clips in May. Natalia sold 48+24 = $\langle\langle 48 + 24 = 72 \rangle\rangle$ 72 clips altogether in April and May. #### 72"

Figure 3: The syntax used in the GSM8K dataset.

In GSM8K, the calculations are explicitly annotated, and removing the tags from chain-of-thought results in natural language sentences. The final result is a single number that is also explicitly annotated at the end of the solution.

We parse the formulas using regular expressions, evaluate them using the SYMPY library (Meurer et al., 2017), and verify that all outputs are numerically close to the values in the data. The conversion into our unified format is a direct one-to-one mapping.

Our analysis shows that the original validation and test splits of GSM8K do not contain duplicates and are not contained in a training split of other datasets.

## 4.2 Ape210K

Ape210K (Zhao et al., 2020) is a dataset of over 200K math problems involving simple arithmetics. The questions are written in Chinese, and the solutions are represented as nested arithmetical expressions and a single numerical result to which they evaluate. We automatically translate the questions to English using Google Translate and linearize the nested expressions into a sequence of simple expressions using depth-first traversal of the expression tree. Figure 4 illustrates the process of linearization.

Furthermore, we discard all examples that cannot be parsed. Then, we evaluate the linear se-

```
Nested expression:
  (2 - 8) + (2 - 8) * (50% + 3)

Linear chain:
  2 - 8 = -6
  50 / 100 = 1/2
  (1/2) + 3 = 7/2
  (-6) * (7/2) = -21
  (-6) * (-21) = -27
```

Figure 4: Linearization of nested expression

quence of steps and remove examples whose end result does not numerically match the original result saved in the data. We also discard all examples with the original result written in the form of "$\langle$number$\rangle$($\langle$fraction$\rangle$)", such as 1(1/2) because of the ambiguity between implicit multiplication and mixed fractions, which are both present in the data in the same form. In total, more than 97% of the examples in each split passed all checks and were kept in the dataset.

Finally, the linearized examples can be directly transformed into our unified format. While Ape210K is much larger than GSM8K, the exported chains do not contain any comments in natural language, and the English prompts are machine-translated.

Analysis of overlaps shows that around 60% of both validation and test examples present duplicates or near-duplicates to the Ape210K's training split. In CALC-X, we remove these examples from validation and test splits with around 1700 remaining in each.

## 4.3 AQuA-RAT

AQuA-RAT (Ling et al., 2017) is a dataset of 100K math problems. The annotations consist of 1) multiple choices, 2) the correct choice, and 3) an informal, free-text rationale that explains the selected choice. The answer is usually a single number but can also be a pair of numbers (coordinates), include a unit, a ratio, "None of the options," and others.

The rationale is in free-text format and generally not parseable with formal grammar. In some cases, calculations are written in words, such as "ten pages per day means seventy pages per week." We approach this in a best-effort manner and use regular expressions to find equations in the form of *expression = number*. We remove all the non-symbolic characters (mainly all textual characters) from both sides of such-identified equations and evaluate the left-hand side using SYMPY calculator.

Finally, we compare the calculator output with the right-hand equation side, and if the result of the calculator matches, we insert a tagged calculator call into the rationale. This results in 1.6 calculator calls on average per single reasoning chain.

Our error analysis shows that the annotators are usually consistent in their rationale structure. The described parsing heuristic works well for the annotations that consistently use "=" (equals symbol) in their chain. However, we usually do not inject any calculator calls for many others that do not follow the equation-following structure. Thus, for applications with high priority of recall in the injected gadget calls, we propose to further filter our dataset to the samples with at least three calculator calls.

Analysis of data leaks shows around 2% of the training split are near-duplicates of around 30% and 25% of test and validation AQuA-RAT samples, respectively. In CALC-X collection, we remove these samples from the train split.

## 4.4 MathQA

MathQA (Amini et al., 2019) is a subset (37K) of AQuA-RAT with further annotations. Human annotators have corrected errors inside the AQuA-RAT rationales and annotated the solution with a nested expression that leads to the correct answer.

We parse the nested expressions and linearize them using a similar procedure as for Ape210K. Less than 0.3% of examples were removed due to parsing or evaluation problems. We also replace all function calls (such as *circle_area*) with corresponding elementary operations that can be executed with a SYMPY calculator.

Next, we keep the examples only if their expression evaluation result is in ±5% range of the selected correct choice in the data, which results in a loss of around 30% of the data. We note that the mismatch of the computed results with annotated options is not consistent with the authors' claim[2] that the expressions in the dataset are guaranteed to evaluate to the selected option, but is consistent with observations by Parisi et al. (2022). After inspection, we attribute most of these errors to the inconsistency in the original MathQA dataset. In our published variant of the dataset, we remove all examples in which the expression does not evaluate to a value numerically close to the selected option.

Evaluation of data leakages shows that *all* samples of MathQA originate from the *training* split

---

[2]math-qa.github.io/math-QA, accessed 20/10/2023

of AQuA-RAT. Hence, we completely remove the validation and test splits of MathQA in CALC-X and omit evaluations on MathQA in our results.

## 4.5 MAWPS

MAWPS (Koncel-Kedziorski et al., 2016) is a collection of around 5000 elementary school-level problems from online websites. The solution to each problem is annotated as an equation with a single variable $x$. Solving the equation for x gives the answer to the problem. We isolate $x$ from the equations by manual annotation and then linearize the corresponding expression into a sequence of calculations to convert the data into our unified format. We do not train our models on any MAWPS data to ensure a fair comparison with previous work.

Around 70% of MAWPS's train samples are near-duplicates of its train split, test split, or ASDiv-A test split. We remove these samples from CALC-X's train collection.

## 4.6 ASDiv-A and SVAMP

ASDiv (Miao et al., 2020) is an arithmetics benchmark with problems of similar difficulty as MAWPS. ASDiv-A picks around 1,200 samples with a number as a solution and a nested expression evaluating to the correct result. SVAMP (Patel et al., 2021a) comprises 1,000 math problems derived from ASDiv, overcoming some of its deficiencies.

Whole ASDiv-A and SVAMP datasets were directly convertible to our common format. With no official train-test split, we use both for evaluation only.

## 5 Experiments

To explore the potential of our newly curated data collection, we train models of identical architecture and parametrization in two configurations:

1. **Train on the original datasets**: use all of the selected datasets (see Section 4) to train a baseline: a generative model that produces an associated output reasoning chain on a given input sequence. All training samples removed from CALC-X are also removed from baseline data for fair comparison.

2. **Train CALCFORMERS on the CALC-X datasets**: train the model for an identical objective, but on the corresponding CALC-X datasets, to demonstrate the interaction with a symbolic

|  |  | GSM8K | AQuA-RAT | Ape210K | MAWPS | SVAMP | ASDiv-A |
|---|---|---|---|---|---|---|---|
| GPT-3 | 175B |  |  |  | 19.9* | 10.0* | 14.0* |
| Toolformer | 6.7B |  |  |  | 44.0* | 29.4* | 40.4* |
| T5-L | 700M | 17.0±3.4 | 26.6±6.6 | 19.0±1.8 | 11.2±2.8 | 18.2±2.3 | 29.7±2.5 |
| Calcformer-T5-L | 700M | **34.2**±4.4 | 27.2±6.6 | **53.1**±2.3 | **39.4**±4.1 | **34.4**±3.0 | **55.6**±2.8 |
| T5-XL | 3B | 19.2±3.6 | 33.5±7.2 | 20.8±1.9 | 16.0±3.1 | 24.7±2.6 | 37.0±2.7 |
| Calcformer-T5-XL | 3B | **39.6**±4.4 | 33.5±6.9 | **53.8**±2.3 | **49.0**±4.3 | **44.7**±3.1 | **66.8**±2.6 |
| Flan-XL | 3B | 24.2±3.8 | 20.8±6.1 | 22.9±2.0 | 15.8±3.1 | 24.0±2.6 | 38.7±2.7 |
| Calcformer-Flan-XL | 3B | **39.4**±4.4 | **31.2**±6.6 | **53.4**±2.3 | **47.1**±4.3 | **46.7**±3.1 | **72.5**±2.5 |

Table 1: **Percentage of correct results** on test sets of listed datasets, for (i) models of previous work, (ii) Calcformer models trained on our Calc-X collection and evaluated with access to sympy calculator, and (iii) language models trained on the original datasets, evaluated in standard sequence-to-sequence generation. Confidence intervals computed in a bootstrapped evaluation (sample size n=500, repeats r=1,000); **bold** denotes significantly best results for a combination of *base model + dataset*. Values marked with * are self-reported results from Schick et al. (2023).

system. In inference, whenever the model generates the enclosing *gadget* tags, the model's generated text is extended with the output of the sympy calculator (see Figure 1).

In both settings, we fine-tune T5 foundation models of Raffel et al. (2020), and Chung et al. (2022) in 700-million and 3-billion parameters' versions, using teacher forcing and cross-entropy loss, commonly applied on sequence-to-sequence Transformers (Bahdanau et al., 2016; Vaswani et al., 2017). We use greedy decoding in inference.

We evaluate both systems by numerically comparing the value of the final result extracted from the generated answer with the ground truth result. In the case of AQuA-RAT, where the answer is one of the options, we compare the predictions against all options and pick the one with the lowest Levenshtein distance as chosen by the model.

In the case of the Calcformer models, the answer is enclosed in the <result> tags that we use to extract the answer. For the baseline models, we extract the answer as the sequence following the key phrase "The final result is"; this format is presented in all the baselines' training samples. Our training setup is further detailed in Appendix A.

**Results** Table 1 compares the performance of the conventional generative models and calculator-supported models trained on Calc-X datasets. Calcformers surpass the accuracy of the baseline models 2-3 times, with the exception of the AQuA-RAT dataset. Nevertheless, the overall improvement of the calculator-using models in reaching the correct answer is 99.6% on average across all datasets.

We can see that on the AQuA-RAT dataset, Calcformers perform comparably with baselines in the case of two out of three base models. We find that this is due to the low average number of tool calls inside the AQuA-RAT training split, leading to inconsistent usage of the calculator on AQuA-RAT test questions.

## 6 Conclusion

This paper introduces a Calc-X dataset collection, transforming over 300,000 samples of arithmetic reasoning datasets into a unified chain-of-thought format with explicit annotation of interaction with a calculator. Calc-X enables integration of a simple symbolic system in the reasoning chains of language models via traditional supervised learning, easily allowing the models to offload mathematical computation to an external tool.

We support the correct use of the Calc-X collection for *both* training and evaluation by unifying the format of all included datasets and eliminating the datasets' mutual data leakages, making Calc-X a convenient default for any future research addressing models' arithmetic reasoning.

Finally, we demonstrate the potential of Calc-X by utilizing the whole unified collection in training and adjusting the models' inference process for the use of the calculator. As a result, our calculator-assisted models reach an accuracy that approximately *doubles* the accuracy of the traditional generation and outperforms existing previous work. We make the Calc-X collection and newly-created calculator-supported Calcformer models publicly available to facilitate further research in the fast-paced area of tool-using language models.

## Limitations

We acknowledge the limitations of our heuristic for injecting annotations into AQuA-RAT rationales. We suggest future authors experiment with utilizing a sequence-to-sequence language model similarly to the method by Schick et al. (2023), which might yield a higher recall.

Further, we note that in some CALC-X datasets, the format of reasoning chains differs from others in that it does not contain verbal explanations surrounding the computational parts. We believe that using a language model here to write explanatory comments in natural language between the computation steps is a promising path if a consistent format of CoT, including both arithmetics and free-text reasoning, is desired.

We also note the limitations of a simple calculator in advanced mathematical reasoning. While the models' extension with a calculator circumvents an important bottleneck, difficult mathematical tasks might require more general symbolic systems to obtain satisfactory results.

## Acknowledgments

This work was supported by the Martina Roeselová Memorial Fellowship granted by the IOCB Tech Foundation. Computational resources were provided by the e-INFRA CZ project (ID:90254), supported by the Ministry of Education, Youth and Sports of the Czech Republic. Further, we acknowledge the Centre for Biomedical Image Analysis at Masaryk University supported by MEYS CR (LM2023050 and CZ.02.1.01/0.0/0.0/18_046/0016045 Czech-BioImaging) for their support with training the models evaluated within this paper. The work has also received funding from the Czech Science Foundation (project number 19-27828X).

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

# A    Training details

## A.1    Model Settings

We trained the models using the HuggingFace Transformers library. We applied standard sequence-to-sequence training with **cross-entropy** loss on all tokens. We optimized the model with **AdamW** optimizer and effective **batch size** of 32. We used **learning rate** of 5e-5 with 1000 **warmup steps**, and a **linear lr decay** to 0 in 400000 steps. The models were trained in bf16 **precision**. All models were trained on a mixture of all datasets, either in CALC-X or in original format, with **data upsampling** to balance different dataset sizes.

## A.2    Training progress

During training, we monitored the percentage of validation predictions that have a correct final result. We compute the performance on each dataset separately and average them together, which we use for **early stopping** and selecting the best checkpoint after training.

A comparison of the models on the aggregate metric during training is illustrated in Figure 5. The detailed (per-dataset) metrics of the best model can be seen in Figure 6.

## A.3    Hardware

To train each of our models, we used a single NVIDIA A100 80GB GPU, 40GB of RAM, and 4 CPU cores. We have trained two models with 700M parameters and four models with 3B parameters, with a total training wall time of around 21 days.

## A.4    Datasets, Models, and Code

We make our code, CALC-X datasets, and models publicly available.

**Code:** https://github.com/prompteus/calc-x

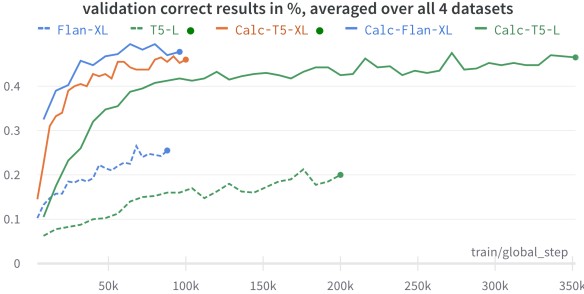

Figure 5: Percentage of validation predictions with correct result during training. Baseline T5-XL run is missing from because of technical difficulties requiring re-runs from intermediate checkpoints.

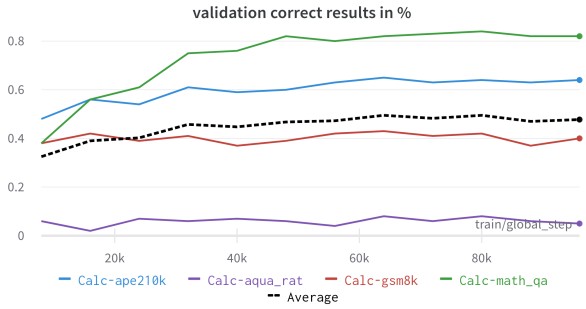

Figure 6: Percentage of validation predictions with a correct result of a single run Calcformer-Flan-XL on each dataset during training.

**Datasets**

- huggingface.co/datasets/MU-NLPC/Calc-math_qa
- huggingface.co/datasets/MU-NLPC/Calc-gsm8k
- huggingface.co/datasets/MU-NLPC/Calc-aqua_rat
- huggingface.co/datasets/MU-NLPC/Calc-ape210k
- huggingface.co/datasets/MU-NLPC/Calc-svamp
- huggingface.co/datasets/MU-NLPC/Calc-mawps
- huggingface.co/datasets/MU-NLPC/Calc-asdiv_a

Please note that more datasets might be added to the CALC-X in the future; see the project repository for an up-to-date list and ready-to-use examples of the full CALC-X collection.

**Calculator-supported Models**

- huggingface.co/MU-NLPC/calcformer-flan-xl
- huggingface.co/MU-NLPC/calcformer-t5-xl
- huggingface.co/MU-NLPC/calcformer-t5-large

See the corresponding model cards for usage.