# OpenReview forum: "Calc-X and Calcformers: Empowering Arithmetical Chain-of-Thought through Interaction with Symbolic Systems"
_EMNLP/2023/Conference — EMNLP 2023 Main_

### Official Review · Reviewer_jPuE · 2023-07-27

**Soundness:** 2

**Excitement:**

3: Ambivalent: It has merits (e.g., it reports state-of-the-art results, the idea is nice), but there are key weaknesses (e.g., it describes incremental work), and it can significantly benefit from another round of revision. However, I won't object to accepting it if my co-reviewers champion it.

**Paper Topic And Main Contributions:**

This short paper addresses the problem of data normalization in chain of thought (CoT) problems.
There currently exists several datasets for CoT and acquiring data for these task is time consuming and expensive.
Furthermore the current data sets are heterogeneous and encoded in different formats, entailing a situation where data preprocessing becomes a real issue.

The authors introduce an HTML/XML format (dubbed Calc-X) for encoding the existing data sets which is easy to process with existing libraries.
They specifically describe the conversion processes for datasets GSM8K,Ape210K,AQuA-RAT and MathQA

There is a short experimental section relying on fine tuning models from the T5 family on the task by comparing performances on the original datasets and on the newly encoded datasets.

**Reasons To Accept:**

- The common formatting and data normalization idea of the paper is surely relevant for the scientific community

**Reasons To Reject:**

- The experiments are not that meaningful and not easily reproducible given the informations provided in the paper:
the comparisons of scores between datasets that end up being different (some examples are dropped by the conversion process) are nonsense to me.
- The authors conclude that they get superior accuracy with their recoded dataset but this claim is not supported by the content
- Although there is a ressources and evaluation track at EMNLP, I feel this paper would be more appopriate in a ressources and evaluation conference like LREC.


**Reproducibility:**

3: Could reproduce the results with some difficulty. The settings of parameters are underspecified or subjectively determined; the training/evaluation data are not widely available.

**Reviewer Confidence:**

3: Pretty sure, but there's a chance I missed something. Although I have a good feel for this area in general, I did not carefully check the paper's details, e.g., the math, experimental design, or novelty.

---

> ### Author Rebuttal · Authors · 2023-08-29
>
> Thank you for reviewing our work and providing us with the feedback. We try to address your outlined Weakness points below.
>
> * **Reproducibility:** Given the space constraints, we try to keep the flow of the main article mostly high-level, but please note that we performed many extra steps to make our work as reproducible as possible, already for the anonymized submission; Complementary to the training details in Appendix A, on the first page we reference a [Huggingface hub](https://huggingface.co/emnlp2023) with both our datasets and models, including instructions for their use. Additionally, in Appendix A, we reference [anonymized GitHub repository](https://github.com/emnlp2023sub/gadgets) with instructions for the exact reproduction of the results reported in Table 1 as well as the training scripts with all parameters intentionally fixed.
>
> * **Data filtering:** We believe there is a miscommunication about the training data of baseline models. Data instanced filtered in the conversion process were removed for *both* calculator-using models *and* baselines. This can be verified with our anonymized GitHub repository [1]. However, we confirm that the paper does not explicitly state this fact, and we *will* include it in the camera-ready version.
>
> * **Claimed superior accuracy:** Our claim about superior accuracy achieved with Calc-X datasets is supported by the results reported in Table 1 - a comparison of *Calc-* models with the corresponding baselines shows that the improvement is two-fold or higher in a majority of cases. However, please note that we do not attribute the improvement to our format directly but primarily to the fact that the **models trained on Calc-X format can use (and do use) a calculator during inference**, eliminating the source of arithmetic errors.
>
> * **Suitability for the Resources track of EMNLP:** Please note that additionally to the proposed data transformations, important contributions of our work are also in a proposition and reproducible implementation of the tool-using model, as well as the empirical verification of efficiency of all the presented components with a variety of datasets and base models.
>
> Please let us know if you still have concerns about any of our discussed claims (reproducibility, data filtering, superior results of Calc-X models) and our proposed improvement for the camera-ready. If the discussion dispersed some of these, we would appreciate it if you give one more consideration to our assigned *Soundness* rating.
>
> -----
>
> [1] GitHub repository: https://github.com/emnlp2023sub/gadgets. Relevant parts: lines 46-49 in file *examples/train\_calc\_balanced\_baseline.py* and definition of *dataset_to_keys* variable in file *examples/baseline\_utils.py*. Note that the baseline's training implementation loads *our* version of the datasets from HuggingFace, which *are already filtered* and contain the data in both ours and the original formatting.

---

### Official Review · Reviewer_yZbJ · 2023-07-31

**Soundness:** 4

**Excitement:**

3: Ambivalent: It has merits (e.g., it reports state-of-the-art results, the idea is nice), but there are key weaknesses (e.g., it describes incremental work), and it can significantly benefit from another round of revision. However, I won't object to accepting it if my co-reviewers champion it.

**Paper Topic And Main Contributions:**

This work focuses on the ability of large language models to use a calculator as a tool. To study this ability, the authors find 4 arithmetic reasoning datasets that have chain-of-thought annotations and convert them into a format (Calc-X) that allows the language model to utilize a calculator in addition to chain-of-thought. The authors consider 3 model variations (T5-large, T5-XL, and FLAN-XL), training each model on the original chain-of-thought dataset as well as their novel Calc-X format. This work demonstrates that through a unified tool-use format, language models can significantly improve accuracy on arithmetic problems.

Overall, I think this paper provides a clear and focused study on arithmetic reasoning within chain-of-thought prompting, and demonstrates the benefits quite well.

**Questions For The Authors:**

Based on my understanding of section 4, it sounds like a few thousand examples from Ape210k, AQuA-RAT, and MathQA were removed because their format couldn’t be parsed by your data annotation method. I think it would be very interesting to see how many of those removed examples can be solved correctly by models trained on the Calc-X format. Do you have results for anything along those lines?

**Reasons To Accept:**

The datasets proposed in this work, as well as the format, may be quite useful for future research into tool-use in LLMs

The results from this study are strong evidence of the benefit of tool use for LLMs for arithmetic reasoning.

**Reasons To Reject:**

Prior works have clearly demonstrated the benefit of calculator use in large language models [1], so it’s a little unclear just how much novelty is derived from the results of this study. However, this work considers different datasets from previous works, and provides a more concentrated study on arithmetic reasoning within chain-of-thought.

[1] Schick et al. Toolformer: Language Models Can Teach Themselves to Use Tools

**Reproducibility:**

4: Could mostly reproduce the results, but there may be some variation because of sample variance or minor variations in their interpretation of the protocol or method.

**Reviewer Confidence:**

4: Quite sure. I tried to check the important points carefully. It's unlikely, though conceivable, that I missed something that should affect my ratings.

---

> ### Author Rebuttal · Authors · 2023-08-29
>
> Thank you for your time and for valuable feedback on our work.
>
> While recent previous works demonstrate the benefit of using a calculator during inference, contrary to these, we make all our datasets, code, and trained models publicly available, making it easier for other researchers to train and work with tool-using models. As we identify in the Related Work section, without standardized supervised datasets, much of the energy in previous work was invested in overcoming the data scarcity; they develop prompting methods with heuristic filtering to generate data [1], utilize self-training [2], or spend resources on proprietary annotation collection [3], all of which are slower, more complex, or more costly methods than training from a supervised dataset.
>
>
> **Question: Performance on dropped-out data**
>
> * **Clarification of data removal:**
>
>     First, we would like to clarify how many data instances are omitted and why.
>
>     * In MathQA, only less than 0.3% of examples were removed due to parsing/syntactical problems. The rest (cca 30%) was removed because of inconsistency in the data, which we observed in agreement with other prior works [2, 4] that inspected the annotation in the MathQA dataset.
>     * In Ape210k, less than 3% of the data was removed, with *ambiguity* being the prevalent cause, not a parser insufficiency. We manually inspected the cause and found that mixed fractions and implicit multiplication were written the same way, making it impossible to distinguish between them algorithmically.
>     * In GSM8K and AQuA-RAT, we did not remove any instances, but we recommend filtering out AQuA-RAT instances with a low number of gadget calls in Section 4.3 and in Limitations.
>
> * **Evaluation on dropped-out data:**
>
>     Evaluating models on dropped-out data is costly because it requires manual human re-annotation - i.e., solving the math problems. However, we performed a small-scale evaluation on the dropped-out samples and report our results below.
>
>     We selected 20 random instances in both MathQA and Ape210k (from the test split) that were dropped during the conversion process, solved them manually, and then manually compared the solutions to outputs from Calc-Flan-XL and Baseline-Flan-XL models.
>     For transparency, we make the manually collected evaluations within this analysis available as notebooks in our anonymized repository [5].
>
>     * From 20 **MathQA**-dropped instances, 1 question was ill-formed with no valid solution. Calc-Flan-XL agreed with our solution in 7 out of the remaining 19 questions, and Baseline-Flan-XL agreed on 0 (none) of the 19 instances. While lower success rates compared to the measures in Table 1 for both models suggest that non-parseable samples are systematically more difficult, we also see that calculator-aided models are more robust in this data segment.
>
>     * From 20 **Ape210k**-dropped instances, 9 contained ambiguity between implicit multiplication and mixed fraction in the question itself (and not just the solution), so we disqualified them from the evaluation. From the remaining 11 problems, 8 were solved correctly by Calc-Flan-XL, which seems consistent with the performance measured in Table 1 on kept instances. Baseline-Flan-XL solved none of the remaining 11 problems correctly.
>
>
> [1] Schick et al.; [Toolformer: Language Models Can Teach Themselves to Use Tools](https://arxiv.org/abs/2302.04761).
>
> [2] Aaron Parisi et al.; [Talm: Tool augmented language models](https://arxiv.org/abs/2205.12255).
>
> [3] Thoppilan et al.; [LaMDA: Language Models for Dialog Applications](https://arxiv.org/abs/2201.08239).
>
> [4] Shen-Yun Miao et al.; [A Diverse Corpus for Evaluating and Developing English Math Word Problem Solvers](https://arxiv.org/abs/2106.15772).
>
> [5] https://github.com/emnlp2023sub/gadgets/tree/master/notebooks

---

### Official Review · Reviewer_ZPay · 2023-08-05

**Typos Grammar Style And Presentation Improvements:** The paper is clear and well-presented.
**Soundness:** 4

**Excitement:**

4: Strong: This paper deepens the understanding of some phenomenon or lowers the barriers to an existing research direction.

**Missing References:**

None that I can think of.

**Paper Topic And Main Contributions:**

The paper contributes a dataset that demonstrates use of calculator tool in mathematical reasoning chains. The dataset consists of more than 300K examples created using GSM8K, Ape-210K, AQuA-RAT, and MathQA, all of which contain mathematical reasoning chains involving calculations. To create the examples, the authors parse the three datasets into a unified HTML-style format, where calculations are expressed as <gadget></gadget> and <output></output>.

Fine-tuning T5 models on this data format instead of the original data leads to clear improvements in % of correct results, although there is no improvement for AQuA-RAT, which appears to also require intermediate computation results.

The authors have already released the model and dataset at anonymous endpoints on Huggingface, with clear documentation.

**Questions For The Authors:**

A: The improvement on datasets like GSM is not large. What are the typical types of error made by the fine-tuned T5 models?

B: Why did you decide to have the model call tools using the HTML tag format? Could this impact the performance?

**Reasons To Accept:**

- The dataset with unified format for arithmetic tool use is valuable, and the strong results show it clearly improves the math reasoning ability of smaller LMs.
- The author have made the dataset and fine-tuned model available, which can facilitate progress in building this capability into LLMs more reliably.
- Clear documentation of the data creation process as well as the released artifacts.

**Reasons To Reject:**

The proposed approach for data format conversion may have limited applicability beyond datasets with simple and well-structured arithmetic operations. As demonstrated by the difficulties with the AQuA-RAT dataset, the proposed processing strategy does not generalize well to more free-form reasoning traces.

**Reproducibility:**

4: Could mostly reproduce the results, but there may be some variation because of sample variance or minor variations in their interpretation of the protocol or method.

**Reviewer Confidence:**

4: Quite sure. I tried to check the important points carefully. It's unlikely, though conceivable, that I missed something that should affect my ratings.

---

> ### Author Rebuttal · Authors · 2023-08-29
>
> Thank you for reviewing our work; we appreciate your questions and feedback.
>
> We agree with your comment on limited applicability beyond datasets that are well-structured to begin with. However, we still identify some possibilities for how future work can benefit from *both* free-form traces and structured Calc-X datasets to create calculator-assisted models:
>
> 1. Pretrain on free-form data and then finetune on Calc-X data to teach the model to use tools.
> 2. Train on a mixture of free-form and Calc-X data and inform the model which output format is expected via a prompt. During inference, always prompt the model to use Calc-X format to encourage tool-using.
> 3. Utilize LLM to rewrite free-form answers into our format by injecting the tags, similar to Schick et al. [1].
>
> **Question 1A:**
> We would argue that improvements in models' performance on the GSM dataset *are* substantial - from 17.0 to 34.2, from 19.2 to 39.6, and from 24.2 to 39.4. In other words, Calc-X models are around twice as likely to arrive at a correct final result than baselines in the GSM dataset.
>
> The typical errors for the baseline models can be divided into two broad categories: (1) in reasoning and (2) in arithmetics. Calc-X models eliminate arithmetic errors as they outsource the computation to a reliable symbolic system. However, they still make errors in reasoning - such as drawing incorrect conclusions in multi-step reasoning or applying irrelevant formulas.
>
> **Question 1B:**
> We choose the HTML-like tag format for its efficient parseability with existing tools, making the resulting models practical for integration in more complex inference scenarios, such as in conversational systems.
> In related settings, previous work utilizing T5 models [2] found that a specific choice of interaction format does not play a substantial role in performance; this makes sense to us as the "learning to interact correctly" task is semantically much simpler than the downstream fine-tuning task at hand.
>
> In line with this assumption, we also observe that models learn to generate valid (i.e., parseable) HTML tags early in the training process, and syntactical errors related to HTML formatting are practically non-existent in the outputs of fine-tuned models. Speculatively, HTML format could help models train faster because HTML syntax structure might appear in the pretraining corpus, but we do not support this hypothesis with quantitative evidence.
>
>
>
> [1] Schick et al.; [Toolformer: Language Models Can Teach Themselves to Use Tools](https://arxiv.org/abs/2302.04761)
>
> [2] Raffel et al.; [Exploring the Limits of Transfer Learning with a Unified Text-to-Text Transformer](https://arxiv.org/abs/1910.10683); Section 2.4.

---

### Meta-Review · Area_Chair_tC6G · 2023-09-17

**Recommendation:** 4

**Metareview:**

The paper introduces a dataset designed to showcase the use of a calculator tool in mathematical reasoning chains. By leveraging four existing datasets, namely GSM8K, Ape-210K, AQuA-RAT, and MathQA, the authors have created over 300K examples. These datasets were parsed into a unified HTML-style format, termed Calc-X, where calculations are represented using specific tags. When T5 models are fine-tuned using this format, there's a noticeable improvement in the accuracy of results, except for the AQuA-RAT dataset. The authors have made the dataset and the fine-tuned model available on Huggingface, accompanied by comprehensive documentation. The paper emphasizes the potential of a unified tool-use format in enhancing the accuracy of language models in arithmetic problems.

The paper's contribution of a dataset with a unified format for arithmetic tool use is invaluable. The results indicate a clear enhancement in the mathematical reasoning abilities of language models. The authors have made both the dataset and the fine-tuned model accessible to the public, which can further stimulate advancements in this domain. The documentation detailing the data creation process and the released resources is commendably clear. The datasets and format proposed in this work could be instrumental for future research on tool-use in large language models. The study's outcomes serve as robust evidence supporting the advantages of tool use for arithmetic reasoning in large language models.

The paper's proposed data format conversion strategy might have limited scope, especially for datasets that involve complex and less structured arithmetic operations. This limitation is evident from the challenges encountered with the AQuA-RAT dataset. While the paper emphasizes the benefits of using calculators in large language models, similar findings have been previously reported, raising questions about the novelty of this study.  A reviewer suggests that the paper might be better suited for a resources and evaluation conference rather than an NLP conference.

---

### Decision · Program_Chairs · 2023-10-07

**Decision:**

Accept-Main

**Comment:**

The paper introduces a dataset designed to showcase the use of a calculator tool in mathematical reasoning chains. By leveraging four existing datasets, namely GSM8K, Ape-210K, AQuA-RAT, and MathQA, the authors have created over 300K examples. These datasets were parsed into a unified HTML-style format, termed Calc-X, where calculations are represented using specific tags. When T5 models are fine-tuned using this format, there's a noticeable improvement in the accuracy of results, except for the AQuA-RAT dataset. The authors have made the dataset and the fine-tuned model available on Huggingface, accompanied by comprehensive documentation. The paper emphasizes the potential of a unified tool-use format in enhancing the accuracy of language models in arithmetic problems.

The paper's contribution of a dataset with a unified format for arithmetic tool use is invaluable. The results indicate a clear enhancement in the mathematical reasoning abilities of language models. The authors have made both the dataset and the fine-tuned model accessible to the public, which can further stimulate advancements in this domain. The documentation detailing the data creation process and the released resources is commendably clear. The datasets and format proposed in this work could be instrumental for future research on tool-use in large language models. The study's outcomes serve as robust evidence supporting the advantages of tool use for arithmetic reasoning in large language models.

The paper's proposed data format conversion strategy might have limited scope, especially for datasets that involve complex and less structured arithmetic operations. This limitation is evident from the challenges encountered with the AQuA-RAT dataset. While the paper emphasizes the benefits of using calculators in large language models, similar findings have been previously reported, raising questions about the novelty of this study.  A reviewer suggests that the paper might be better suited for a resources and evaluation conference rather than an NLP conference.